# Kinetic Study of BLV Infectivity in BLV Susceptible and Resistant Cattle in Japan from 2017 to 2019

**DOI:** 10.3390/pathogens10101281

**Published:** 2021-10-05

**Authors:** Lanlan Bai, Liushiqi Borjigin, Hirotaka Sato, Shin-Nosuke Takeshima, Sakurako Asaji, Hiroshi Ishizaki, Keiji Kawashima, Yuko Obuchi, Shinji Sunaga, Asako Ando, Hidehito Inoko, Satoshi Wada, Yoko Aida

**Affiliations:** 1Photonics Control Technology Team, RIKEN Center for Advanced Photonics, 2-1 Hirosawa, Wako 351-0198, Saitama, Japan; lanlan.bai@riken.jp (L.B.); takesima@jumonji-u.ac.jp (S.-N.T.); swada@riken.jp (S.W.); 2Viral Infectious Diseases Unit, RIKEN, 2-1 Hirosawa, Wako 351-0198, Saitama, Japan; liushiqi.borjigin@vetmed.hokudai.ac.jp (L.B.); hirosato@dokkyomed.ac.jp (H.S.); 3Department of Food and Nutrition, Jumonji University, 2-1-28 Sugasawa, Niiza 352-8510, Saitama, Japan; 4GenoDive Pharma Inc., 4-14-1 Naka-cho, Atsugi 243-0018, Kanagawa, Japan; sasaji@genodive.co.jp (S.A.); aando@is.icc.u-tokai.ac.jp (A.A.); hinoko@genodive.co.jp (H.I.); 5Grazing Animal Unit, Division of Grassland Farming, Institute of Livestock and Grassland Science, NARO, 768 Senbonmatsu, Nasushiobara 329-2793, Tochigi, Japan; hishizak@affrc.go.jp; 6Tobu and General Agricultural Office Livestock Hygiene Division, Ota 373-0805, Gunma, Japan; kawashima-ke@pref.gunma.lg.jp; 7Department of Agriculture Dairy and Livestock Division, Maebashi 371-8570, Gunma, Japan; obuchi-yu@pref.gunma.lg.jp (Y.O.); sunaga-shi@pref.gunma.lg.jp (S.S.); 8Department of Molecular Life Science, Division of Basic Medical Science and Molecular Medicine, Tokai University School of Medicine, 143 Shimokasuya, Isehara 259-1119, Kanagasa, Japan; 9Laboratory of Global Infectious Diseases Control Science, Graduate School of Agricultural and Life Sciences, The University of Tokyo, 1-1-1 Yayoi, Bunkyo-ku, Tokyo 113-8657, Japan

**Keywords:** BLV, syncytium formation, proviral load, *BoLA-DRB3* allele, horizontal transmission, susceptible, resistant

## Abstract

Bovine leukemia virus (BLV) is the causative agent of enzootic bovine leukosis. Polymorphism in bovine lymphocyte antigen *(BoLA)-DRB3* alleles is related to susceptibility to BLV proviral load (PVL), which is a useful index for estimating disease progression and transmission risk. However, whether differential *BoLA-DRB3* affects BLV infectivity remains unknown. In a three-year follow-up investigation using a luminescence syncytium induction assay for evaluating BLV infectivity, we visualized and evaluated the kinetics of BLV infectivity in cattle with susceptible, resistant and neutral *BoLA-DRB3* alleles which were selected from 179 cattle. Susceptible cattle showed stronger BLV infectivity than both resistant and neutral cattle. The order of intensity of BLV infectivity was as follows: susceptible cattle > neutral cattle > resistant cattle. BLV infectivity showed strong positive correlation with PVL at each testing point. BLV-infected susceptible cattle were found to be at higher risk of horizontal transmission, as they had strong infectivity and high PVL, whereas BLV-infected resistant cattle were low risk of BLV transmission owing to weak BLV infection and low PVL. Thus, this is the first study to demonstrate that the *BoLA-DRB3* polymorphism is associated with BLV infection.

## 1. Introduction

Bovine leukemia virus (BLV), an oncogenic member of the genus *Deltaretrovirus* of family *Retroviridae*, is the etiological agent of enzootic bovine leukosis (EBL), the most common neoplastic disease in cattle [1,2]. Most cattle infected with BLV are asymptomatic, and approximately one-third develop persistent lymphocytosis, characterized by nonmalignant polyclonal CD5^+^ B-cell expansion, while only a small percentage develop EBL after a long latency period. Currently, vaccines or therapeutic procedures for preventing BLV transmission are lacking, because of which, the virus has spread worldwide. EBL was also added to the International Organization of Epizootics (OIE)-listed diseases, infections, and infestations in 2020 [3]. Currently, 40.9% of dairy cows over 6 months of age and 28.7% of breeding cattle in Japan [4], 94.2% of dairy herds in America [5], 36.7% of cattle and 78.3% of herds in Canada [6], over 81.8% of cattle and 99.1% of herds in Taiwan [7], over 50% of cattle and 86.8% of herds in Korea [8], approximately 31–41.9% of cattle in China are BLV seropositive [9,10]. BLV infection induces huge economic losses in cattle production and export [11] as it affects herd levels in high-performing dairy herds and cow longevity [12,13]. Reports show that BLV infection leads to significant economic losses in the dairy industry in the USA, which is estimated to be USD 525 million annually [11]. In addition, the economic loss per case of lymphosarcoma due to BLV was estimated to be USD 412 to USD 497 (CAD 635) in recent studies [13,14].

The free virus is unstable, and herd infection occurs mainly via BLV-infected cells. Horizontal transmission is recognized as the major route of BLV infection [15]. Either a small volume of blood with infected lymphocytes or a thousand infected cells are sufficient to infect a healthy animal [16]. BLV infection can be an iatrogenic disease, as it can be contracted during dehorning, ear tattooing, rectal palpation, reuse of needles and gloves [17,18], and vectors [19,20]. In addition, insects and seasonal variations (the highest infection rates occur from July to September and the lowest infection rates occur during winter) are closely associated with the spread of BLV infection under natural conditions [20]. BLV can also infect newborns via the placenta during pregnancy and colostrum feeding from BLV-infected cows [21,22]. Binding of the BLV envelope to cationic amino acid transporter 1 [23], a cellular receptor of BLV, mediates cell fusion and virus entry. The BLV genome integrates into the host genome as a proviral load (PVL), which correlates strongly with disease progression [24,25] and BLV infectivity, assessed via syncytium formation [26]. Sato et al. developed a new assay called the luminescence syncytium induction assay (LuSIA), which is based on a new reporter cell line called CC81 (a feline cell line transformed by mouse sarcoma virus)-GREMG. This cell line can specifically respond to the expression of BLV regulatory protein, Tax, when cultured together with BLV-infected cells to form enhanced green fluorescent protein (EGFP)-expressing syncytium [26], which enables visualization of BLV cell-to-cell infection in vitro. Reports show that when PVLs exceed 10,000 copies/10^5^ cells in blood, cattle secrete BLV in nasal secretions, saliva [27], and milk [28]. Furthermore, BLV from milk shows infectivity [28]. Therefore, BLV-infected cattle with high PVL are sources of infection for BLV-free cattle. Thus, PVL is considered a major diagnostic index for estimating BLV transmission risk [29].

PVL is strongly associated with the highly polymorphic bovine leukocyte antigen *(BoLA)-DRB3* [30,31,32]. In total, 384 alleles are registered in the Immuno Polymorphism Database (IPD)-MHC database (https://www.ebi.ac.uk/ipd/mhc/group/BoLA/, accessed on 2 September 2021). The *BoLA-DRB3*015:01* and *DRB3*012:01* alleles are known susceptibility-associated markers related to high PVL. In contrast, the *BoLA-DRB3*009:02*, *DRB3*014:01:01* [30,31,33,34], and *DRB3*002:01* alleles [33] are resistant markers associated with the development of high PVL. Other *BoLA-DRB3* alleles were not significantly associated with PVL in vivo [30]. Thus, *BoLA-DRB3* alleles are believed to determine cattle–specific differences resistant to BLV disease progression. However, whether polymorphisms in *BoLA-DRB3* alleles are related to BLV infectivity is not known.

In present follow-up study spanning three years, we assessed whether different *BoLA-DRB3* alleles were associated with BLV infectivity using white blood cells (WBCs) from BLV-infected cattle with different *BoLA-DRB3* alleles and PVL.

## 2. Results

### 2.1. Determination of BoLA-DRB3 Alleles and BLV Infection

Previously, we had reported that *BoLA-DRB3*015:01* and *DRB3*012:01* alleles are associated with high PVL of BLV. The *BoLA-DRB3*009:02*, *DRB3*014:01:01* and *DRB3*002:01* alleles were associated with high PVL resistance [30,31,33,34], while other *BoLA-DRB3* alleles were not significantly associated with PVL in vivo [30]. For selecting BLV-infected cattle for three years (from 2017 to 2019) of follow-up investigation, we first collected blood from all cattle (*n* = 179) from Farm A in Tochigi prefecture, Japan, and analyzed the *BoLA-DRB3* alleles and BLV infection. DNAs were extracted from blood samples to be typed *BoLA-DRB3* allele using sequence-based typing (SBT) method. Twenty alleles of *BoLA-DRB3* in the IPD-MHC database were identified in our samples. As shown in Figure 1A, the results of allele frequency showed that 16.0% resistant alleles, 22.0% susceptible alleles, and 62.0% neutral alleles were detected in 179 cattle. The rate of resistant *BoLA-DRB*3 allele is known to be lesser than that of other alleles [31,35]. On the basis of the *BoLA-DRB3* allele data, 179 cattle were grouped into three cattle groups: resistant cattle, susceptible cattle, and neutral cattle, which were defined to be carrying at least one susceptible, resistant or neutral allele in their genomes, respectively (Table 1).

Next, we measured the PVL of known and novel BLV variants in BLV-infected animals using BLV-CoCoMo-qPCR-2 (RIKEN Genesis, Kanagawa, Japan) [25,36,37,38]. In total, 142 out of 179 cattle (79.3%) were positive and 37 (20.7%) were negative for BLV PVL (Table 1). The average PVL of each cattle group showed that the mean PVL of 49 resistant cattle was 4216 copies per 10^5^ cells, that of 62 susceptible cattle was 19,206 copies per 10^5^ cells, and that of 68 neutral cattle was 14,350 copies per 10^5^ cells (Figure 1B). Furthermore, the PVLs of resistant cattle were significantly higher than those of susceptible cattle (*p* = 0.00004) and neutral cattle (*p* = 0.00669) (Figure 1B). In contrast, 16 resistant cattle, 4 susceptible cattle, and 17 neutral cattle were BLV-negative (Table 1).

In particular, the BLV provirus can be detected in milk, nasal mucus, and saliva samples of dairy cattle with PVL of more than 10,000, 14,000, and 18,000 copies/10^5^ cells in blood samples, respectively [27,28]. Therefore, we set a threshold between high- and low-PVL to 10,000 copies/10^5^ cells, as described previously [33]. Based on the threshold of PVL, 62 susceptible cattle, 68 neutral cattle, and 49 resistant cattle were divided into three groups: high PVL, low PVL, and BLV-free (Table 1). The range of PVL of 142 cattle was 14 to 70,870 copies per 10^5^ cells. The 35 susceptible cattle (56.5%) had high PVL, with mean PVL of 32,218 copies/10^5^ cells, ranging from 10,154 to 65,564 copies per 10^5^ cells, while the 23 susceptible cattle (37.1%) had low PVL, with mean PVL of 2746 copies per 10^5^ cells, ranging from 9 to 9761 copies per 10^5^ cells. In contrast, only six resistant cattle had high PVL and the mean PVL was 28,688 copies/10^5^ cells, ranging from 12,984 to 39,614 copies per 10^5^ cells, and the remaining 27 resistant cattle (55.1%) had low PVL and the mean PVL was 1275 copies/10^5^ cells, ranging from 14 to 8122 copies per 10^5^ cells (Table 1). In addition, as shown in Figure 1B, the PVLs in almost all resistant cattle were lower than the threshold (10,000 copies/10^5^ cells).

Taken together, the high infection rate in susceptible cattle with high PVL was compared to that in neutral and resistant cattle. Our results revealed the presence of BLV-infected resistant and susceptible cattle with different levels of PVLs, which were sufficient for the three-year follow-up study.

### 2.2. Syncytium Formation Abilities in Cattle with Different BoLA-DRB3 Alleles

PVL is a risk factor of BLV infection that correlates with BLV infectivity and is evaluated based on the syncytium formation assay [25,26]. Next, to select target BLV-infected cattle for the three-year follow-up investigation, we randomly tested 24 out of 142 cattle to analyze BLV infectivity using LuSIA [39] in the second half of 2017 (Table 2). Anti-BLV antibodies were detected in serum obtained from all tested blood samples. The 24 BLV-infected cattle contained 15 cattle carrying a susceptible *BoLA-DRB3*015:01* or *DRB3*012:01* allele, eight cattle carrying a resistant *BoLA-DRB3*009:02* or *DRB3*014:01:01* allele, and one cattle carrying a neutral allele. The tested cattle with different levels of PVL were assessed using CoCoMo-qPCR-2. All susceptible cattle (S1-15) had high PVLs that exceeded 10,000 copies/10^5^ cells, one neutral cattle N1 also carried 12,423 copies/10^5^ cells, and all resistant cattle (R1-7) had low PVLs of less than 10,000 copies/10^5^ cells. The WBCs obtained from the tested blood samples were co-cultured with the BLV reporter cell line CC81-GREMG, which can respond to Tax expression to express green fluorescence [39]. Nuclei were stained with Hoechst 33,342, which showed blue fluorescence. As shown in Figure 2A, susceptible cattle (S2, S3, and S7) had large syncytia that expressed EGFP, while resistant cattle (R2, R4, and R5) showed small number of syncytia. We compared the syncytium formation ability of all 24 tested cattle (Figure 2B). All resistant cattle, except R1 (8597 copies per 10^5^ cells), showed obviously weaker BLV infectivity than all susceptible cattle and one neutral cattle, as shown in Figure 2B. The results showed that BLV syncytium formation ability was weak in resistant cattle and strong in susceptible cattle.

### 2.3. Assessing BLV Infectivity in Seven Selected Cattle in Three-Year Follow-Up Study

In this three-year follow-up investigation, we would not constrain any decisions of the farmer as depositary breeding, selling, slaughtering, and eliminating et al. At the beginning, we decided to select 13 out of 24 cattle for BLV infectivity syncytium formation assay, but some selected cattle were sold out or deposited with other vendors’ deposit breeding services, while at the time of our sampling. Therefore, as shown in Table 2, only 7 out of 13 cattle were sampled at each checking time owing to some selected cattle were sold out (S10 sold out after the second time sampled) or deposited with other vendors’ deposit breeding services, while at the time of our sampling. These 7 BVL-infected cattle contained three susceptible cattle (S2, S3, and S7), three resistant cattle (R2, R4, and R5), and one cattle with the neutral allele, N1. From October 2017 to August 2019, we collected blood samples from targeted cattle seven times and assessed BLV infectivity using LuSIA [39]. The syncytium was formed via cell fusion after BLV infection, and hence syncytium formation ability reflected BLV infectivity. The WBCs were isolated from each blood sample and syncytium formation assay was performed using the BLV reporter cell line CC81-GREMG at each time point. The permanently BLV-infected FLK-BLV cell line was used as a positive control. EGFP-expressing syncytia were observed using an EVOS2 fluorescence microscope. The syncytia index (SI) of syncytium indicated the relative syncytium number of each sample when syncytium number in FLK-BLV cells was set to 1. As shown in Figure 3, the SI of syncytium in resistant cattle R2, R4, and R5 was consistently lower than that in susceptible cattle S2, S3, and S7 at each checking point in the follow-up period, indicating that BLV infectivity was weak in resistant cattle but strong in susceptible cattle. The neutral cattle N1 also showed medium BLV infectivity in the three cattle groups. Taken together, the results showed that resistant cattle had weak BLV infectivity, which did not increase throughout the long post-infection period. In contrast, susceptible cattle showed strong BLV infection during their infectious phase. Our results indicated that resistant cattle indeed have low risk of BLV transmission, and that selection of these resistant cattle represents a promising approach for controlling the spread of the virus.

### 2.4. Correlation of BLV Infectivity and PVL in Three-Year Follow-Up Study

*BoLA-DRB3* alleles are associated with PVL in vivo. In addition, our three-year follow-up study clearly showed that BLV infectivity was weak in resistant cattle, strong in susceptible cattle, and medium in neutral cattle N1. However, the association of the *BoLA-DRB*3 allele with BLV syncytium formation ability is unknown. Therefore, to clarify the relationship between *BoLA-DRB3* allele and BLV infectious ability, we increased the number of tested cattle to 13, which the selected cattle in the depositary breeding service returned to the farm (Table 2), and then evaluated BLV infectivity using LuSIA and PVL using BLV-CoCoMo-qPCR-2 seven times during our follow-up study. We considered the effects of uncontrollable external factors on the infectivity of BLV and PVL at each sampling time; thus, we analyzed the data detected at each checking time for separate analyses. The PVL and number of syncytia of tested cattle at each checking point were represented seven times, as shown in Figure 4A. With the exception of sample R1, the susceptible cattle showed higher PVLs (blue spot) and larger number of syncytia (orange bar) than those in neutral cattle (gray bar) and resistant cattle (green bar). Results of previous reports [24,33], have shown that the *BoLA-DRB3*009:02* allele suppressed PVL better than the *BoLA-DRB3*014:01:01* allele. Here, the resistant cattle R1 carried the *BoLA-DRB3*014:01:01* allele, and had high PVL and a large number of syncytia. Furthermore, to analyze the correlation between PVL and syncytium at each checking point, we constructed a scatter graph and performed linear regression analysis using the data at each time point for separate analyses. Interestingly, a strong positive correlation was observed, and the Spearman’s rank correlation coefficient (R) ranged from 0.7611 to 0.9350 (R = 0.8537 ± 0.0716) (Figure 4B). This is also consistent with our previous results [39]. Taken together, our findings demonstrated that resistant cattle showed lower PVL and weaker BLV infectivity in their long infectious period than susceptible cattle.

## 3. Discussion

Among the many risk factors associated with BLV infection, *BoLA* complex polymorphisms are one of the most important host factors strongly involved in controlling the subclinical progression of BLV infection by regulating PVL in vivo [24,27,28,33,40,41,42,43,44], as PVL is an indicator of disease progression. Several studies have shown that *BoLA-DRB3* is a highly polymorphic gene that affects susceptibility to BLV-induced B cell lymphoma [45,46], and PVL [31]. BLV infection mainly occurs via cell-to-cell transmission in individuals and herds. Based on the results of a three-year follow-up study, we collected blood from seven BLV-infected cattle, including three susceptible cattle (*BoLA-DRB3*015:01* or *DRB3*012:01*), three resistant cattle (*BoLA-DRB3*009:02* or *DRB3 *014:01:01*), and one neutral cattle, and showed for the first time that *BoLA-DRB3* alleles are associated with BLV infectivity.

*BoLA-DRB3*016:01* and *DRB3*015:01* have been reported to be associated with high PVL in Japanese Black and Holstein cattle, respectively [30,45]. All BLV-susceptible cattle tested also had high PVLs, and WBCs from their blood showed strong BLV infectivity in a follow-up investigation from 2017 to 2019. In contrast, *BoLA-DRB3*009:02* is known to play an important immunological role in suppressing viral replication, resulting in resistance to disease progression in both Japanese Black and Holstein cattle [47]. Cattle R2 and R5 carry resistant *BoLA-DRB3*009:02* allele, and they had weak BLV infectivity and low PVL in the long-infected phase. In addition, no BLV transmission was observed over 30 months of contact when cattle carrying resistant *BoLA-DRB3*009:02* allele with low PVL were incorporated into a BLV-negative dairy herd. The *BoLA-DRB3*0902* cattle with low PVL disrupted the BLV transmission chain [29]. Our results are consistent with previously reported that BLV-infected susceptible cattle have high PVL and remain at its high level, and resistant cattle keep low PVL over a long infection period [33,48]. Thus, resistant cattle be considered that are low risk of BLV transmission.

The *BoLA-DRB3*014:01:01* has also been reported to suppress both BLV PVL and lymphoma [33]. The tested resistant cattle, with the exception of cattle R1, showed lower PVL and weaker BLV infectivity than susceptible cattle in three years. In the initial phase of the follow-up study, resistant cattle R1 had higher PVL and infectivity than the other tested resistant cattle, despite carrying the *BoLA-DRB3*014:01:01* allele. Although *BoLA-DRB3*014:01:01* suppressed PVL to some extent in vivo, infection suppression was possibly not completely achieved. The PVL of R1 increased and exceeded the threshold value of 10,000 copies/10^5^ cells. Besides, the *BoLA-DRB3*002:01* allele [33] has been reported that is a resistant marker associated with high PVL. The nearly 20% (6 of 33) BLV-infected resistant cattle had high PVL carry *DRB3*014:01:01* allele or *DRB3*002:01* allele. Thus, it is considered that resistant *DRB3*009:02* allele can strongly suppress the development of high PVL than *DRB3*014:01:01* allele or *DRB3*002:01* allele. It may be that *DRB3*009:02* combines with other host factors to thoroughly suppress high PVL.

In Japanese Black herds, the distribution of allele frequencies for *BoLA-DRB3*016:01* was over 30%, *DRB3*009:02* was 2.0~5.8%, and *DRB3* 014:01:01* was 1.4~2.5%. In Holstein cattle herds, the distribution of allele frequencies for *BoLA-DRB3*015:01* was 19.4~ 35.4%, *DRB3*009:02* was 5.2~7.1%, *DRB3* 014:01:01* was 2.4~7.1%, and *DRB3*002:01* was 4.5~8.6% [45]. The distribution of *BoLA-DRB3* allele frequencies in our selected Farm A also was in the above distribution ranges. Thus, Farm A can be a representative of Japanese farms, and it can interpret Japanese farms. Besides, a strong positive correlation was observed between BLV infectivity and PVL. Therefore, our results are considered that can reflect the state of the entire farm.

R cattle have lower PVL and weak BLV infectivity than susceptible cattle, indicating that R cattle are low risk of BLV transmission. The amount of PVL increased or changed over time. Resistant and susceptible cattle with the same amount of PVL were not detected in the tested cattle at any time point. Therefore, it was difficult to directly compare the direct association of the *BoLA-DRB3* allele with BLV infectivity. However, two major hypotheses were considered. One was that the resistant allele was involved in eliminating BLV infections to reduce its original integration; therefore, PVL did not increase. The other was that resistant alleles regulate viral replication, which lowers PVL production, resulting in weaker BLV infectivity than in susceptible cattle. The resistant *BoLA-DRB*3 allele directly or indirectly affects the BLV infectivity to reduce BLV transmission. In addition, the levels of anti-gp51 antibodies differ significantly in resistant and susceptible cattle [33], indicating that BoLA MHC Class II molecules present different antigen epitopes for T and B cells in these cattle to activate the adaptive and humoral immune response. Resistant cattle may have strong immunity and can protect from BLV.

Although some resistant cattle are infected with BLV, they maintain low levels of PVL due to their long infectious period. Furthermore, a considerable proportion of resistant cattle are BLV-negative. In contrast, most susceptible cattle are easily infected with BLV and have high PVLs in the short post-infection period. Previous reports have shown that the BLV provirus may be detected in milk, nasal mucus, and saliva of dairy cattle with PVLs > 10,000, 14,000, and 18,000 copies per 10^5^ cells in blood samples, respectively [28]. In addition, BLV from raw milk is infectious [28]. Our tested susceptible cattle, with the exception of S14, had high PVL, which was more than the threshold value of 10,000 copies/10^5^ cells. The range of PVL in S14 was also 5274–16,532 copies/10^5^ cells. Therefore, we believe that BLV-infected susceptible cattle might produce BLV in their secretions and milk. BLV-infected susceptible cattle were found to have a significantly higher risk of horizontal transmission. Similarly, vertical transmission risk and horizontal transmission appeared to be extremely high for dams and cattle with susceptible alleles compared to those with resistant alleles [49]. In contrast, the resistant *BoLA-DRB3* allele can effectively inhibit or prevent BLV infection, as observed in this study. Thus, BLV-infected resistant cattle are low risk of BLV transmission because of weak BLV infection with low PVL. In addition, PVL was maintained at low level in most dams with resistant alleles, thereby reducing the risk of vertical BLV transmission [49]. In contrast, BLV-infected susceptible cattle are at a high risk of BLV infectivity. Therefore, in addition to eliminate and removal BLV transmission in high-risk susceptible cattle, breeding of low-risk resistant cattle is considered a promising strategy to gradually reduce the infection rate while minimizing economic loss. Furthermore, based on our understanding of the evolution of the *BoLA-DRB3* allele in cattle, the development of breeding strategies aimed at improving resistance to infectious diseases and designing of broadly effective vaccines against susceptible cattle should be considered in the future.

## 4. Materials and Methods

### 4.1. Blood Sample Collection

In Japan, the number of cattle raised per farm is 93.9. We selected an entire herd of 179 cattle containing Holstein cattle, Japanese Black, and F1 hybrids, which is about twice as large as the average scale. Blood samples were collected from 179 cattle of Farm A in Tochigi Prefecture, Japan, and stored in ethylenediaminetetraacetic acid (EDTA). Serum was collected to detect BLV antibodies. This study was approved by the animal ethical committee, and the animal care and use RIKEN animal experiments committee (Approval Number H29-2-104) and the animal care committee of the Institute of Livestock and Grassland Science, NARO (Approval Number: 1711B082, 1811B084, 1911B041).

### 4.2. DNA Extraction

DNAs were extracted using the Wizard^®^ Genomic DNA Purification Kit (Promega corporation, Madison, WI, USA) at each time point from the blood samples according to the manufacturer’s instructions.

### 4.3. Detection of BLV Proviral Load

BLV PVL was measured in all DNA samples from blood using the BLV-CoCoMo-qPCR-2 method (RIKEN Genesis, Kanagawa, Japan) as described previously [24,25,37,38,50,51].

### 4.4. BoLA-DRB3 Allele Typing

*BoLA-DRB3* alleles were genotyped using the polymerase chain reaction (PCR)-sequence-based typing (SBT) method as described previously [52]. Briefly, primers F (5′-CGCTCCTGTGAYCAGATCTATCC-3′) and R (5′-CACCCCCGCGCTCACC-3′) were used for cDNA amplification of *BoLA-DRB3* exon 2 using PCR. The PCR products were purified to sequence using the Big Dye Terminator v1.1 Cycle Sequencing Kit. The sequence conditions were as follows: 25 cycles at 96 °C for 10 s, 50 °C for 5 s, and 60 °C for 2 min. Sequence data were analyzed using ASSIGN 400 ATF software (Conexio Genomics, Fremantle, Australia) to identify the *BoLA-DRB3* alleles. Susceptible cattle identified as carrying at least one susceptible *BoLA-DRB3* allele, resistant cattle carrying at least one resistant *BoLA-DRB3* allele, and neutral cattle carrying other *BoLA-DRB3* alleles in their genome.

### 4.5. Measurement of gp51 Antibodies

Anti-BLV gp51 antibodies in all serum samples were measured using an anti-BLV antibody ELISA kit (JNC, Tokyo, Japan), according to the manufacturer’s instructions.

### 4.6. Detection of Syncytium Formation

The blood samples were treated with red blood cell lysis buffer (Abbott Diagnostics Technologies AS, Oslo, Norway) to remove red blood cells, and the pellets were washed with cold phosphate-buffered saline (PBS) and then resuspended in Dulbecco’s modified Eagle’s medium (DMEM) (Thermo Fisher Scientific, Waltham, MA, USA) with 10% fetal bovine serum (FBS; Sigma-Aldrich, St. Louis, MO, USA) [53]. WBCs (1 × 10^5^ cells/well) were applied to the BLV reporter cell lines, CC81-GREMG (5 × 104 cells/well), [23,26,39,54] that were pre-cultured for one day in a 12-well plate. After 3 days of incubation, the plate was washed with PBS, and fresh DMEM with 10% FBS was added and incubated for up to 48 h. The cells were washed with PBS and fixed with PBS containing 3.7% formaldehyde and 10 mg/mL Hoechst 33,342 (Millipore Sigma). The fixed cells were observed for fluorescent-positive syncytia using an EVOSFL Auto 2 Cell Imaging System (Thermo Fisher Scientific). The permanently BLV-infected FLK-BLV cell line [55,56] was used as a positive control for this assay.

### 4.7. Statistical Analysis

The correlation coefficient (R) between the syncytium and PVL was calculated using Excel with the Pearson function. Analysis of variance followed by Tukey’s’ post-hoc test was used to determine the significance of the means of PVL for multiple comparisons. Differences were considered to be significant at *p* < 0.05, and strongly significant at *p* < 0.01 and *p* < 0.001.

## 5. Conclusions

We measured BLV infectivity in WBCs via syncytium formation assay among susceptible, resistant, and neutral cattle at three years of follow-up. The susceptible cattle carrying the *BoLA-DRB3*015:01* or *DRB3*012:01* allele showed strong infectivity and high PVL in their blood. Resistant cattle carrying the *BoLA-DRB3*009:02* or *DRB3*014:01:01* allele maintained weaker BLV infectivity and lower PVL at each checking point than the susceptible and neutral cattle. Although we did not directly compare BLV infectivity in the same amount of PVL of susceptible cattle and resistant cattle in vitro, the *BoLA-DRB3* allele was found to be directly or indirectly associated with BLV infectivity. Consequently, breed selection based on resistant *BoLA-DRB3* allele is an effective strategy for reducing and controlling BLV infection.

## Figures and Tables

**Figure 1 pathogens-10-01281-f001:**
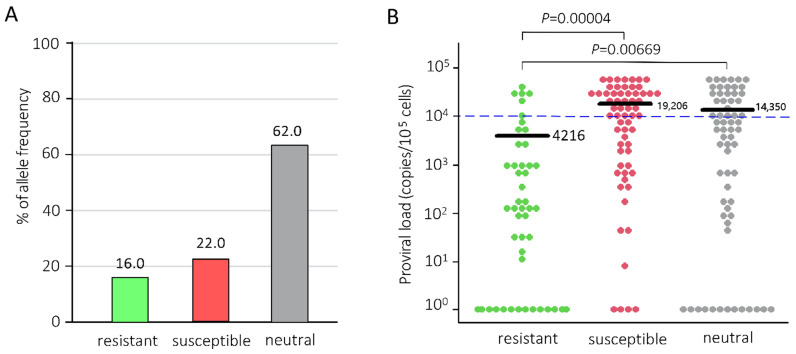
*BoLA-DRB3* allele frequencies (**A**) and estimation of proviral loads (PVLs) (**B**). Blood samples were obtained from all 179 cattle and DNAs were extracted. *BoLA-DRB3* alleles were typed using the SBT method, and the PVLs were measured using the CoCoMo-qPCR-2 method. All cattle were divided into resistant, susceptible, and neutral cattle groups based on the *BoLA-DRB3* allele. The blue dotted line represents a BLV PVL of 10,000 copies/10^5^ cells, which was set as the threshold between high and low PVL. The *X*-axis shows cattle classification, and the *Y*-axis shows percentage of allele frequency (**A**) and proviral loads (**B**). The mean PVL was compared among three groups and *p* value was calculated using Tukey’s test after the analysis of variance. *p* < 0.05 represents statistically significant and 0.05 < *p* < 0.01 represents statistically highly significant results.

**Figure 2 pathogens-10-01281-f002:**
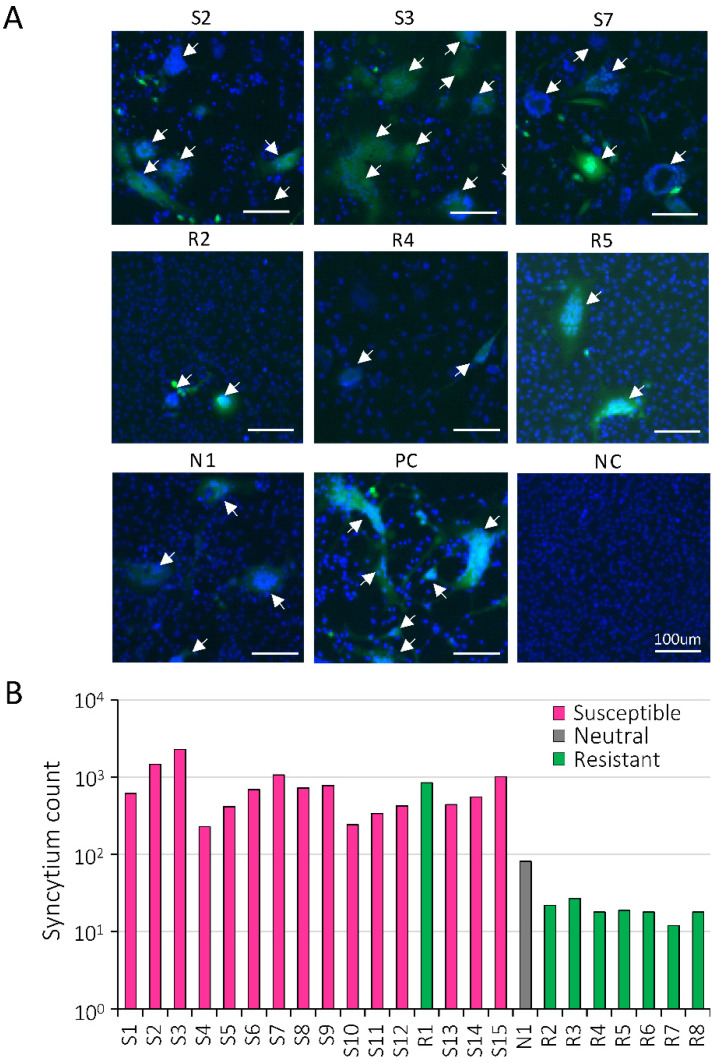
BLV infectivity of white blood cells (WBCs) from BLV-susceptible, -resistant, and -neutral cattle. (**A**) Syncytium formation. (**B**) Comparison of syncytium formation ability. WBCs were isolated from blood of BLV-infected susceptible cattle (S; *n* = 15), resistant cattle (R; *n* = 8), and one neutral cattle (N), and then co-cultured with the BLV reporter cell line, CC81-GREMG, which responded to BLV Tax expression for five days to form an EGFP-expressing syncytium. The fluorescent syncytia were observed using an EVOS2 fluorescence microscope. BLV-infected fetal lamb kidney (FLK-BLV) cells permanently infected with BLV were used as the positive control. Mock-treated cells were used as the negative control. White arrows show EGFP-expressing syncytia in (**A**). The *Y*-axis shows syncytium count, and the *X*-axis shows cattle number.

**Figure 3 pathogens-10-01281-f003:**
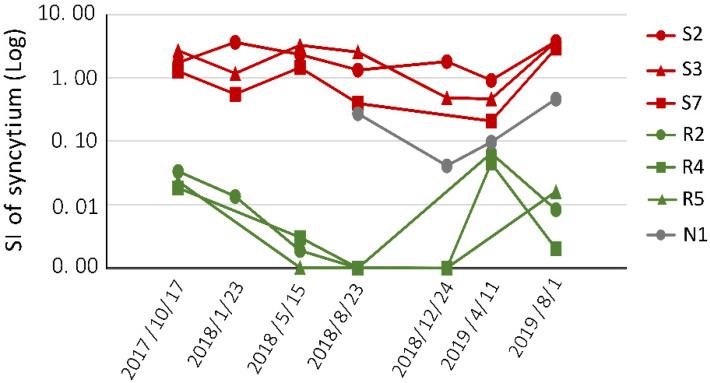
Kinetics of BLV infectivity in BLV-susceptible, -resistant, and -neutral cattle. White blood cells (WBCs) were isolated from the blood of BLV-infected BLV-susceptible (red), -resistant cattle (green), and neutral cattle (grey), and then co-cultured with the BLV reporter cell line, CC81-GREMG, for five days. The fluorescent syncytia were observed using an EVOS2 fluorescence microscope. Permanently infected FLK-BLV cells were used as the positive control. The SI of syncytium was calculated from the proportion of syncytium number that was formed in sample WBCs, while the positive control was set at one.

**Figure 4 pathogens-10-01281-f004:**
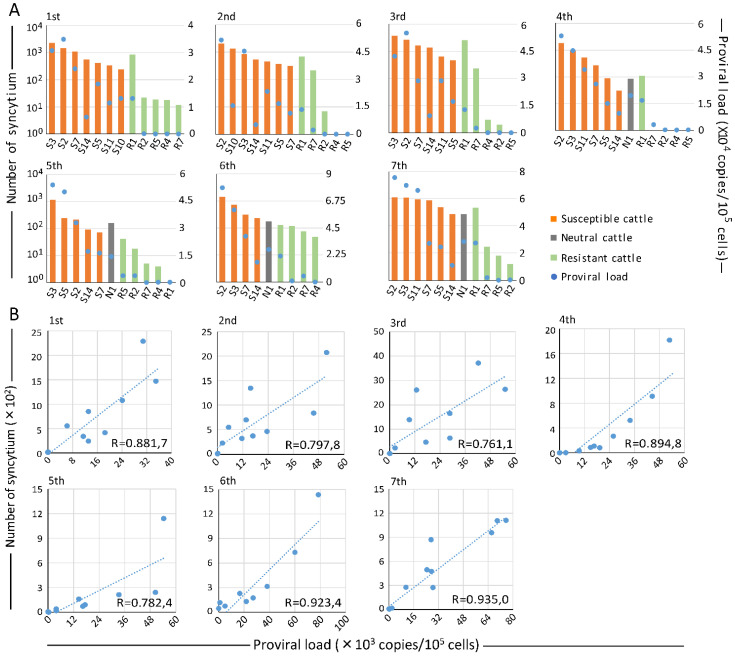
Correlation between BLV infectivity and proviral loads (PVLs) in followed-up cattle. (**A**) Syncytium number and proviral load of tested cattle at each checking point. (**B**). Correlation between syncytium formation ability and PVL at each checking point. During blood sampling, DNAs were extracted and the PVLs were measured using CoCoMo-qPCR-2. The syncytium detected in the white blood cells of BLV-infected cattle were co-cultured with CC81-GREMG cells for five days. Fluorescent syncytia were observed using an EVOS2 fluorescence microscope. BLV infectivity was indicted by the number of syncytia in the bar graph. The bold line represents the approximate curve (R = correlation coefficient).

**Table 1 pathogens-10-01281-t001:** Distribution of cattle according to *BoLA-DRB3* allele and PVL.

Cattle Classification	Cattle No.	High PVL ^1^ Group		Low PVL Group		BLV-Free
		BLV+ (%)	Mean PVL	BLV+ (%)	Mean PVL	BLV− (%)
Susceptible cattle ^2^	62	35 (56.5%)	32,218	23 (37.1%)	2746	4 (6.4%)
Neutral cattle ^3^	68	26 (38.2%)	34,634	25 (36.8%)	3014	17 (25.0%)
Resistant cattle ^4^	49	6 (12.2%)	28,688	27 (55.1%)	1275	16 (32.7%)
Total	179	67 (37.4%)	32,839	75 (41.9%)	2306	37 (20.7%)

^1.^ PVL: proviral load (copies/10^5^ cells). ^2.^ Susceptible cattle carrying at least one susceptible *BoLA-DRB3* allele in their genome. ^3.^ Resistant cattle carrying at least one resistant *BoLA-DRB3* allele in their genome. ^4.^ Neutral cattle carrying other *BoLA-DRB3* alleles.

**Table 2 pathogens-10-01281-t002:** Information on BLV-infected cattle for assessing BLV infectivity and correlation analysis.

Cattle No.	Genotype	*BoLA-DRB3*		PVL ^1,2^	gp51 Abs ^2^	Age ^2^ (Year)	Syncytium Formation Assay for Three-Year Follow-Up Study	Correlation Analysis of BLV Infectivity with PVL
		A	B					
S1	Susceptible	012:01	015:01	79,519	+ ^3^	3.7	NT ^4^	NT
S2	Susceptible	001:01	015:01	35,758	+	7.8	selected	Tested
S3	Susceptible	011:01	015:01	37,707	+	6.2	selected	Tested
S4	Susceptible	011:01	015:01	10,780	+	3.9	NT	NT
S5	Susceptible	001:01	015:01	27,642	+	10.7	NT	Tested
S6	Susceptible	011:01	015:01	36,149	+	7.1	NT	NT
S7	Susceptible	001:01	015:01	39,024	+	11.2	selected	Tested
S8	Susceptible	001:01	015:01	17,981	+	5.8	NT	NT
S9	Susceptible	011:01	015:01	40,534	+	5.1	NT	NT
S10	Susceptible	015:01	015:01	18,062	+	6.8	NT	Tested
S11	Susceptible	015:01	018:01	7733	+	4.8	NT	Tested
S12	Susceptible	011:01	012:01	18,879	+	6.2	NT	NT
S13	Susceptible	010:01	015:01	12,097	+	1.4	NT	NT
S14	Susceptible	010:01	015:01	10,169	+	6.3	NT	Tested
S15	Susceptible	015:01	015:01	16,337	+	6.1	NT	NT
N1	Neutral	001:01	010:01	12,424	NT	0.5	selected	Tested
R1	Resistant	011:01	014:01:01	8597	+	6.2	NT	Tested
R2	Resistant	009:02	015:01	222	+	4.8	selected	Tested
R3	Resistant	001:01	014:01:01	165	+	10.8	NT	NT
R4	Resistant	014:01:01	014:01:01	0	+	6.2	selected	Tested
R5	Resistant	001:01	009:02	0	+	7.8	selected	Tested
R6	Resistant	011:01	014:01:01	54	+	5.4	NT	NT
R7	Resistant	011:01	014:01:01	468	+	5.1	NT	Tested
R8	Resistant	011:01	014:01:01	0	+	4.3	NT	NT

^1.^ PVL: proviral loads (copies/10^5^ cells). ^2.^ Sampled to detect PVL and gp51 antibodies and calculate age in the second half of 2017. ^3.^ Seropositive. ^4.^ NT: Not tested. Susceptible cattle carrying at least one susceptible *BoLA-DRB3* allele in their genome. Resistant cattle carrying at least one resistant *BoLA-DRB3* allele in their genome. Neutral cattle carrying other *BoLA-DRB3* alleles.

## Data Availability

The data presented in this study are available on request from the corresponding author.

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
