# Peer review of "Kinetic Study of BLV Infectivity in BLV Susceptible and Resistant Cattle in Japan from 2017 to 2019"

_pathogens, 2021, doi:10.3390/pathogens10101281_

Round 1

Reviewer 1 Report

In the current study, authors conducted a three-year follow-up study to assess the association of different BoLA-DRB3 alleles with BLV infectivity. The kinetics of BLV infectivity was evaluated by means of CoCoMo-qPCR-2 and also LuSIA. This study demonstrated a strong positive correlation of BLV infectivity with PVL. Moreover, the BoLA-DRB3 polymorphism is associated with BLV infection.

This study has been well planned and the main goal was accomplished by appropriate methods LuSIA and CoCoMo-qPCR-2, despite a small scale of the 3-year follow-up study.

Major comments:

  1. Please describe the criteria for selection of the 24 (out of 179) cattle for the assay in Table 2. Was it based on PVL levels, age, random, etc.? Also, among the 24 cattle, why only 7 samples were enrolled for infectivity syncytium formation assay, and 13 for correlation analysis of BLV infectivity with PVL?

Moreover, my concern is whether the small number of samples can represent the whole herd should be discussed.  

  1. Figure 2: larger size of syncytial foci was observed in BLV-resistant sample (R5), and small syncytial loci also can be found in sensitive samples (S2, S3) in the LuSIA assay. If additional to the overall number of syncytial foci, the size of syncytial foci may also indicate the infectivity, then increase of sample size, particularly the resistance animals, will make this study more convincible.
  2. Table 2 summarized the information of both section 2.2 and 2.3 (Figure 2 and Figure 3). Since both Fig 2B and Fig 3 included results of syncytium formation ability assay; Fig 2B with all 24 samples, while Fig. 3 included only 7 samples enrolled for the long term follow-up experiment. However, in Table 2, the cattle number referred for syncytium formation ability was only “7”. It will be confused. It could be nice if author can keep a note on “syncytium formation ability” in the Table 2 to indicate those samples (7 cattle) are for “three-year follow-up study”.
  3. In section 2.3., it referred “7 out of 24 cattle to assess BLV infectivity in the three-year follow-up study (Figure 3).” What is the criteria for the selection? I am curious about R1, since it induced a higher syncytium formation ability. It will be interesting to follow it up for 3 years. Apparently, accordingly to Table 2, there are 13 samples were selected to the follow-up study for the correlation of PVL and also syncytium formation ability (as shown in Fig 4). Since the 13 samples were subjected to the syncytium formation ability, and therefore I suggested the 13 samples, rather than 7, can all included for Fig 3. Moreover, not all the selected 13 cattle were included each time points in Figure 4; why were different the samples and various total numbers analysed at the different time assay? Please clarify the study design.

  1. Fig 3: Based on syncytium formation, it seems the BLV Infectivity decreased at some time points in the resistant samples. As we known BLV infection remains proviral state; once infected, it remains in the blood. The possible mechanism contributing to the decreased infectivity is worthy of discussion.

Minor:

  1. Table 1: Since all the samples were collected from Farm A, I find it is ambitus to use the term “herd” in this table.

Should “Herd” change to “Cattle number”, and herd (%) in the two PVL groups change to BLV+ (%), while BLV- (%) in the BLV free group.

  1. The use of English can be further refined: such as threshold pointàthreshold; and 2.2 “using” the syncytium formation assayà “based on”

Author Response

Pathogens

Author's Response To Reviewer Comments

Answer to the comments of reviewer #1:

Thank you very much for the helpful and constructive comments. We have amended the manuscript in accordance with these comments and our point-by point responses are set out below.

Comments and Suggestions for Authors

In the current study, authors conducted a three-year follow-up study to assess the association of different BoLA-DRB3 alleles with BLV infectivity. The kinetics of BLV infectivity was evaluated by means of CoCoMo-qPCR-2 and also LuSIA. This study demonstrated a strong positive correlation of BLV infectivity with PVL. Moreover, the BoLA-DRB3 polymorphism is associated with BLV infection.

This study has been well planned and the main goal was accomplished by appropriate methods LuSIA and CoCoMo-qPCR-2, despite a small scale of the 3-year follow-up study.

Major comments:

Please describe the criteria for selection of the 24 (out of 179) cattle for the assay in Table 2. Was it based on PVL levels, age, random, etc.? Also, among the 24 cattle, why only 7 samples were enrolled for infectivity syncytium formation assay, and 13 for correlation analysis of BLV infectivity with PVL?

Answer: In response to reviewer comment, we randomly selected 24 (out of 179) cattle from BLV-infected herd in Table 2. In this three-year follow-up investigation, we would not constrain any decisions of the farmer as depositary breeding, selling, slaughtering, and eliminating et al. At the beginning, we decided to select 13 out of 24 cattle for infectivity syncytium formation assay and correlation analysis of BLV infectivity with PVL, but some selected cattle were sold out or deposited with other vendors' deposit breeding services, while at the time of our sampling, so only 7 out of 13 cattle were sampled at each checking point. Therefore, only 7 samples were enrolled for infectivity syncytium formation assay. When the cattle in the depositary breeding service returned to the farm, they were sampled again for correlation analysis of BLV infectivity with PVL. We have added this information in line 198-205 in the revised manuscript.

Moreover, my concern is whether the small number of samples can represent the whole herd should be discussed. 

Answer: In response to reviewer comment, in pervious study (Lewin, H.A. et al., Anim Genet 1986; Takeshima, S.N. et al., Retrovirology 2019; Hayashi, T. et al., J Vet Med Sci 2017), subclinical progression of BLV infection is under the control of the BoLA-DRB3 complex, the susceptible cattle have high PVL and resistant cattle have low PVL during their infectious phase. Furthermore, our selected Farm A is about twice the average scale of Japanese farm has 34.6% (62/179) susceptible cattle and 27.4% (49/179) resistant cattle. Our results consistent with previously reported that BLV-infected susceptible cattle have high PVL and remain at its high level, and resistant cattle keep low PVL over a long infection period. Besides, a strong positive correlation was observed between BLV infectivity and PVL. Therefore, these results are considered that can reflect the state of the entire farm. We have added these explanations into line 288-294 in “Discussion” section in the revised manuscript.

  In addition, we have added an information about average scale of farm in Japan in line 348-350 in “Materials and Methods” section in the revised manuscript.

Figure 2: larger size of syncytial foci was observed in BLV-resistant sample (R5), and small syncytial loci also can be found in sensitive samples (S2, S3) in the LuSIA assay. If additional to the overall number of syncytial foci, the size of syncytial foci may also indicate the infectivity, then increase of sample size, particularly the resistance animals, will make this study more convincible.

Answer: We agree with the reviewer comment that the size of syncytial foci may also indicate the infectivity, then increase of sample size (Sato H. et al., Arch Virol, 2018, 163, 1519–1530; Virology Journal, 2020, 17, 57). In this study, we only analyzed number of syncytia, did not measure the size of syncytia. In next further study, we also will analyze the size of syncytial foci for BLV infection ability.

Table 2 summarized the information of both section 2.2 and 2.3 (Figure 2 and Figure 3). Since both Fig 2B and Fig 3 included results of syncytium formation ability assay; Fig 2B with all 24 samples, while Fig. 3 included only 7 samples enrolled for the long term follow-up experiment. However, in Table 2, the cattle number referred for syncytium formation ability was only “7”. It will be confused. It could be nice if author can keep a note on “syncytium formation ability” in the Table 2 to indicate those samples (7 cattle) are for “three-year follow-up study”.

Answer: We apologize about our description makes reviewer confusing, we have modified the “Selected for syncytium assay” to “Syncytium formation assay for three-year follow-up study” and “Tested for correlation analysis of BLV infectivity with PVL” to “Correlation analysis of BLV infectivity with PVL” in Table 2 in the revised manuscript.

In section 2.3., it referred “7 out of 24 cattle to assess BLV infectivity in the three-year follow-up study (Figure 3).” What is the criteria for the selection? I am curious about R1, since it induced a higher syncytium formation ability. It will be interesting to follow it up for 3 years. Apparently, accordingly to Table 2, there are 13 samples were selected to the follow-up study for the correlation of PVL and also syncytium formation ability (as shown in Fig 4). Since the 13 samples were subjected to the syncytium formation ability, and therefore I suggested the 13 samples, rather than 7, can all included for Fig 3. Moreover, not all the selected 13 cattle were included each time points in Figure 4; why were different the samples and various total numbers analyzed at the different time assay? Please clarify the study design.

Answer: In response to reviewer, as explain to the first comment, we have selected 13 out of 24 cattle, but only 7 cattle were sampled at each checking time owing to some selected cattle were sold out (S10 sold out after the second time sampled) or deposited with other vendors' deposit breeding services, while at the time of our sampling. When the cattle in the depositary breeding service returned to the farm, they were sampled again for correlation analysis of BLV infectivity with PVL. Therefore, only 7 samples in Figure 3 were sampled at the same time, and the additional samples were analyzed in Figure 4 were sampled at different time points. So that Figure 3 was including 7 samples, and Figure 4 was including different the samples and various total numbers.

The case R1 was described in Lines 280-284 in “Discussion” section in the original manuscript. In the initial phase of the follow-up study, resistant cattle R1 had higher PVL and infectivity than the other tested resistant cattle, despite carrying the BoLA-DRB3*014:01:01 allele. Although BoLA-DRB3*014:01:01 suppressed PVL to some extent in vivo, infection suppression was possibly not completely achieved. In addition, we have added the sentence “the PVL of R1 increased and exceeded than the threshold value of 10,000 copies/105 cells” in line 300-301 in the revised manuscript.

Fig 3: Based on syncytium formation, it seems the BLV Infectivity decreased at some time points in the resistant samples. As we known BLV infection remains proviral state; once infected, it remains in the blood. The possible mechanism contributing to the decreased infectivity is worthy of discussion.

Answer: We agree with the reviewer’s comment that BLV infection remains proviral state; once infected, it remains in the blood. In Figure 3, the Y-axis, SI of syncytium was shown by Log to better show the different of BLV infectivity between susceptible and resistant cattle. The range of change of SI of syncytium in resistant cattle is within 0.1, and it is considered that is not a big change.

Minor:

Table 1: Since all the samples were collected from Farm A, I find it is ambitus to use the term “herd” in this table. Should “Herd” change to “Cattle number”, and herd (%) in the two PVL groups change to BLV+ (%), while BLV- (%) in the BLV free group.

Answer: As reviewer points out, we have changed the “Herd” to “Cattle No.”, “Herd (%)” to “BLV+(%) or BLV-(%)” in Table 1 in the revised manuscript.

The use of English can be further refined: such as threshold pointàthreshold; and 2.2 “using” the syncytium formation assayà “based on”

Answer: Thank you so much for your suggestion, we have changed “threshold point” to “threshold”; and “using the syncytium formation assay” to “based on syncytium formation assay” in the 2.2 section in the revised manuscript.

Reviewer 2 Report

Dear Authors,

The manuscript need extensive language edit!

Comments

Line 53 did you mean dairy cows or  daily cows

Also paragraph 54-59 is very poor structure and grammar

Line 64, the reference “13” must be updated, it is too old 2003

Line 70, change blood sucker to vectors

Line 88 what do you mean by “PVL susceptibility”

Line 103: Previously, we have shown that BoLA-DRB3*015:01 and DRB3*012:01

You did not show, these statement from previous studies, rephrase the sentence

Line 107 years not yeasr

Line 108 to line 110, should be deleted, it is already in methods

Discussion is very poor and did not interpret the results in relation to previous studies. Also, most of obtained results did not clarified the reason

Line 326 you should delete the sentence “DNAs were extracted from  the blood samples to detect PVLs and BoLA-DRB3 alleles: from sample section

Line 344, author should describe the sequencing process and how evaluate the sequence

Author Response

Pathogens

Author's Response To Reviewer Comments

Answer to the comments of reviewer #2:

Thank you very much for the helpful and constructive comments. We have amended the manuscript in accordance with these comments and our point-by point responses are set out below.

Comments and Suggestions for Authors

The manuscript need extensive language edit!

Answer: As the reviewer suggestion, we have used Editage (www.editage.com) for English language editing.

Comments

Line 53 did you mean dairy cows or daily cows

Answer: Thank you so much for your points out, we apologize our misspelling. We have changed “daily” to “dairy” in line 53 in the revised manuscript.

Also paragraph 54-59 is very poor structure and grammar

Answer: As the reviewer suggestion, we have changed the structure and grammar in “submitted paragraph 54-59” to “revised paragraph 54-57” in this manuscript.

Line 64, the reference “13” must be updated, it is too old 2003

Answer: As reviewer points out, we have uploaded another reference 14 “Alessa, K. et al., J Dairy Sci 2019, 102, 2578-2592” in line 62 in the revised manuscript.

Line 70, change blood sucker to vectors

Answer: As the reviewer points out, we have changed “blood sucker” to “vectors” in line 68 in the revised manuscript.

Line 88 what do you mean by “PVL susceptibility”

Answer: In response to reviewer comment, we have changed “PVL susceptibility” to “PVL” in line 86 in the revised manuscript.

Line 103: Previously, we have shown that BoLA-DRB3*015:01 and DRB3*012:01. You did not show, these statement from previous studies, rephrase the sentence

Answer: As reviewer points out, we have changed “we have shown that …” to “we had reported…” in line 101 in the revised manuscript.

Line 107 years not yeasr

Answer: As reviewer points out, we apologize our misspelling, and change “yeasr” to “years” in line 105 in the revised manuscript.

Line 108 to line 110, should be deleted, it is already in methods

Answer: In response to reviewer comment, the description of “results” is before “methods” in the submission format of Pathogens, so we think that the explanation better be described here as what experiments have been done. Therefore, this part is not removed from the results.

Discussion is very poor and did not interpret the results in relation to previous studies. Also, most of obtained results did not clarified the reason

Answer: In response to reviewer comment, we have added more discussion in line 282-294 and line 300-308 in the revised manuscript

Line 326 you should delete the sentence “DNAs were extracted from the blood samples to detect PVLs and BoLA-DRB3 alleles: from sample section

Answer: As reviewer’s comment, we have deleted the “DNAs were extracted from the blood samples to detect PVLs and BoLA-DRB3 alleles” from sample section in the revised manuscript.

Line 344, author should describe the sequencing process and how evaluate the sequence

Answer: In response to reviewer’s comment, we have added the description of sequencing process in line 369-371 in the revised manuscript. The evaluation of sequence was referred by reference 51.

Author Response

Pathogens

Author's Response To Reviewer Comments

Answer to the comments of reviewer #3:

Thank you very much for the helpful and constructive comments. We have amended the manuscript in accordance with these comments and our point-by point responses are set out below.

Statement to the Authors:

Thank you for this interesting and valuable manuscript. I believe this work contributes to the field of BLV research by building upon current understanding about genetic influences on BLV disease and infectivity, which influence transmission risk. For example, the manuscript presents further evidence for PVL as a proxy measure for infectivity by associating PVL with an in vitro assay for syncytium formation, and reinforces the important role of genetics (BoLA-DRB3 alleles) in individual animal susceptibility and resistance to BLV disease.

My main concerns upon review of this manuscript are that the Materials and Methods section seems to be lacking detail, particularly around herd information (which I feel is highly important in a genetics study) and the statistical methods (animal selection for follow-up testing, group sizes, etc.), and that the Results section appeared to have some inconsistencies (or points where I was confused enough to perceive inconsistencies). I would also like to see the Discussion section expanded in the areas of study limitations and future directions – this desire is primarily related to my concerns regarding the Methods (group sizes, etc.). I also feel the Conclusions section (and related section of the Abstract) should be refined to better align with the study scope, goals, and results.

Some refinement of the wording would improve the clarity of this manuscript. Two areas where this would make the most impact are in relation to using terminology consistently and being clear about whether the authors are referring to findings from this study or from the literature. In regards to terminology, the terms infection, infectivity, and transmission can have similar, but subtly different meanings, but in some cases seems to be used interchangeably in the text. I feel the authors would be well served to ensure their meaning is clear when using these terms, particularly in the Discussion section.

Specific additional suggestions are below, arranged by section:

Abstract

In parallel with my comments regarding the Conclusions section, I feel that the claim that resistant cattle“were not a source of infection” (Line 38) is not supported by a study that did not examine this outcome. The first half of this sentence is focused around transmission risk, which I think is more appropriate, although the wording could again be refined so as not to imply that transmission of BLV was an outcome measured in this study.

Answer: As reviewer’s suggestion, we have changed “BLV-infected resistant cattle were not a source of infection…” to “BLV-infected resistant cattle were low risk of BLV transmission… ” in line 38 in the revised manuscript.

  • Line 33: sentence wording is slightly confusing – perhaps insert “which” (…alleles which were selected from…”

Answer: As reviewer’s suggestion, we have inserted a word “which” to become “alleles which were selected from 179 cattle” in line 33 in the revised manuscript.

Introduction

The introduction provides good background and references, explaining the basics of the virus, disease, prevalence, and impact on industry.

  • Line 52: I suggest you remove the word “been”.

Answer: As reviewer’s suggestion, we have removed the word “been” from the sentence to become “EBL was also added to the International Organization of Epizootics…” in line 52 in the revised manuscript.

  • Line 53-59: I typically expect % to be followed by “of” (x percent of y) – I would recommend the addition of this word throughout. I also suggest the authors double check how they have summarized the referenced studies; for example, the LaDronka study [5] reported that 94.2% of dairy herds contained at least one BLV positive cow by milk ELISA, however, the current wording of the sentence reads as though the study found 94.2% of dairy cows were seropositive, which is both incorrect and (to get a little technical) misrepresents the testing method.
  • Line 53 and 55: I believe “daily cows” was intended as “dairy cows”
  • Line 54: I suggest “are” instead of “show”

Answer: As reviewer’s suggestion, we have added “of” after %. As long as one cow in the herd was BLV positive, the herd was classified as BLV positive. Therefore, we have changed the structure and grammar in “submitted paragraph 54-59” to “revised paragraph 54-57” in this manuscript.

   Thank you so much for the reviewer points out, we apologize our misspelling. We have changed “daily” to “dairy” in line 53 in the revised manuscript.

As reviewer’s suggestion, we use “are” instead of “show” in line 57 in the revised manuscript.

  • Line 59-64: While I agree that BLV results in economic loss, both the Ott [11] and Rhodes [13] are from 2003 – I feel that “a recent study” is not an accurate statement and would suggest investigating whether any newer economic studies or tools are available related to the costs of BLV, or at least indicating that these numbers are nearly two decades old.

Answer: As reviewer points out, we have uploaded another reference 14“Alessa, K. et al., J Dairy Sci 2019, 102, 2578-2592”in line 62 in the revised manuscript.

  • Line 67: I find this sentence slightly confusing – a small volume of blood contains thousands of infected cells, so I’m not sure what the “or” is referring to.

Answer: In response to reviewer comment, a thousand of lymphocyte is equivalent to the number of lymphocytes in approximately 0.1 microliter blood. In here, we want to explain “A small volume of blood with infected lymphocytes can infect a healthy animal” or “A thousand infected cells are also sufficient to infect a healthy animal”. Therefore, we have modified our sentence to “Either a small volume of blood with infected lymphocytes or a thousand infected cells a sufficient to infect a healthy animal” in line 64-66 the revised manuscript.

  • Line 68: I suggest indicating that BLV “can be” an iatrogenic disease rather than “is”, since this is not the only means of transmission and the authors follow this statement by describing natural transmission methods.

Answer: As reviewer points out, we have changed “BLV infection is an iatrogenic disease” to “BLV infection can be an iatrogenic disease” in line 66 in the revised manuscript.

  • Line 74: envelope (instead of “envelop”)

Answer: Thank you so much for the reviewer points out, we apologize our misspelling. We have changed “envelop” to “envelope” in line 72 in the revised manuscript.

  • Line 75: suggest removing “is”

Answer: As reviewer’s suggestion, we have removed “is” from “is a cellular receptor of BLV” to be “a cellular receptor of BLV” in line 73 the revised manuscript.

Results

I found the results interesting but challenging; interesting because of the informative results but challenging because there does not seem to be a clear explanation for how and why results from certain cattle were analyzed in the various sections and not others, and this is not addressed in the Materials and Methods.

I am also somewhat confused by Table 2 and Figures 2 and 4. In Table 2, column 8 is titled “Tested for correlation analysis of BLV infectivity with PVL” however, several of the “tested” cattle in this column are also listed as “NT” or “not tested” in the syncytium formation assay. For example, S14 is listed as NT for the syncytium formation assay, yet syncytium results for S14 are present in both Figure 2(B) and 4(A). In addition, Table 2 shows information for 23 cattle; Line 163 indicates there are 8 R cattle, but the table only shows information for R1-R7, while results for R8 are present in Figure 2(B) but not in Figure 4(A).

Answer: We apologize about our description makes reviewer confusing, we have modified the “Selected for syncytium assay” to “Syncytium formation assay for three-year follow-up study” and “Tested for correlation analysis of BLV infectivity with PVL” to “Correlation analysis of BLV infectivity with PVL” in Table 2 in the revised manuscript.

   As reviewer points out, we apologize to omit the information of R8 in Table 2. We have added the information of R8 in the revised Table 2.

   In response to reviewer’s comment, in this three-year follow-up investigation, we would not constrain any decisions of the farmer as depositary breeding, selling, slaughtering, and eliminating et al. At the beginning, we decided to select 13 out of 24 cattle for infectivity syncytium formation assay and correlation analysis of BLV infectivity with PVL, but some selected cattle were sold out or deposited with other vendors' deposit breeding services, while at the time of our sampling. Therefore, only 7 (S2, S3, S7, N1, R2, R4, R5) out of 13 cattle were sampled at each checking time owing to some selected cattle were sold out (S10 sold out after the second time sampled) or deposited with other vendors' deposit breeding services, while at the time of our sampling. When the cattle in the depositary breeding service returned to the farm, they were sampled again for correlation analysis of BLV infectivity with PVL. We have added this information in line 198-205 in the revised manuscript. Besides, cattle R8 did not select for detecting BLV infectivity for three-year follow-up and correlation analysis of BLV infectivity with PVL.

I find it interesting that, from the visual representation in Figure 3, it appears that R and N cattle experienced more variation in SI from measurement to measurement. I would be interested to know if the authors feel this is an artifact of the small sample size (could be discussed here) or may indicate something that warrants further investigation, particularly in terms of potential risk to the success of BLV control programs (more suitable for the Discussion, perhaps).

Answer: In response to reviewer comment, in Figure 3, the Y-axis, SI of syncytium was shown by Log to better show the different of BLV infectivity between susceptible and resistant cattle. The range of change of SI of syncytium is within 0.1 in resistant cattle (R) and 0.04-0.46 in neutral cattle (N), indicating that are not big variation. In addition, the amount of PVL in resistant and neutral cattle did not have big changes.

  • Line 105: BLV (infection? Disease?) resistance or high PVL resistance? The paper is framed around PVL – suggest keeping this consistent as much as possible.

Answer: As reviewer points out, we have changed “BLV resistance” to “high PVL resistance” in line 103 in the revised manuscript.

  • Line 107: years (instead of ‘yeasr’)

Answer: As reviewer points out, we apologize our misspelling, and change “yeasr” to “years” in line 105 in the revised manuscript.

  • Line 146: suggest indicating 142 BLV-infected cattle and dropping “out of 179”

Answer: As reviewer points out, we have dropped out “out of 179” to be “The range of PVL of 142 cattle was 14 to 70,870 copies per 105 cells” in line 144 in the revised manuscript.

  • Line 163: again, suggest indicating 142 BLV-infected cattle rather than referencing the full herd

Answer: As reviewer points out, we have correlated the sentence to “we tested 24 out of 142 cattle to analyze BLV infectivity using LuSIA” in line 161 in the revised manuscript.

  • Line 168: the preceding alleles are identified, and “the resistant… allele” seems to imply the same will be true here; perhaps “a resistant… allele” would suffice?

Answer: As reviewer points out, we have modified “the susceptible… allele” to “a susceptible… allele”, “the resistant… allele” to “a resistant… allele”, and “the neutral allele” to “a neutral allele” in line 164-166 in the revised manuscript.

  • Line 209: suggest defining SI when first used

Answer: As reviewer suggestion, we have identified “SI is syncytia index” in line 214 in the revised manuscript.

  • Line 226: SI (instead of IS)

Answer: Thank you so much for pointing out, we have correlated “IS” to “SI” in line 230 in the revised manuscript.

  • Line 245: cattle R1 had a PVL of 8,597 which was below the cutoff for the authors’ definition of“high PVL” – please clarify.

Answer: In response to reviewer, cattle R1 had a PVL of 8,597 which was below the cutoff (10,000 copies/105 cells) at the beginning of the test when figure 2 experiment was performed. The PVL of cattle R1 increased and exceeded our cutoff from the second sampling time. We also have added this information in line 300-301 in the revised manuscript. Therefore, we explained as “The resistant cattle R1 carried the BoLA-DRB3*014:01:01 allele, and had high PVL and a large number of syncytia” in the submitted manuscript

Discussion

I would like to see some additional discussion about study limitations and future directions. For example, only one herd was sampled in this study and the number of animals followed was relatively small. How generalizable are the conclusions of this study, in the view of the authors? Do the authors know if the allele distribution for susceptibility, neutral, and resistance (16-62-22) is typical for Japanese herds?

Answer: As reviewer’s suggestion, we have added some additional discussion in line 288-294 in the revised manuscript. Our research group also do about allele distribution in Japanese herd (Lo C.W et al, Viruses 2020, 12, 352; Pathogens 2021, 10, 437; Takeshima S.N. et al, Retrovirology 2019, 16, 14; Miyasaka T. et al, Gene 2011, 472, 42–49).

In a recent paper (Hutchinson, 2021 doi: 10.3390/pathogens10080987), PVL was shown to remain relatively stable over time, although individual animals sometimes experienced large variations in PVL, which agrees with this study’s findings. I would be curious to know the authors’ thoughts on how important, then, the nearly 20% (6 of 33) BLV-infected resistant cattle which had high PVL, or the increased variation in SI that appears to occur in R cattle could be to the success of BLV control programs.

Answer: We agree with the reviewer’s comment. While PVL reaching a certain high level that exceeds our cutoff, PVL is shown to remain relatively stable over time.

In response to reviewer comment, BoLA-DRB3*009:02 is known to strongly suppress viral replication, resulting in resistance to disease progression in both Japanese Black and Holstein cattle. In contrast, DRB3*014:01:01 can suppress PVL to some extent in vivo, infection suppression was possibly not completely achieved. The resistant cattle R1 carrying the BoLA-DRB3*014:01:01 allele has higher PVL and infectivity than the other tested resistant cattle. Furthermore, the BoLA-DRB3*002:01 allele (Lo, C.W. et al., Viruses 2020, 12, doi:10.3390/v12030352.) has been reported that is a resistant marker associated with high PVL. The nearly 20% (6 of 33) BLV-infected resistant cattle had high PVL carry DRB3*014:01:01 allele or DRB3*002:01 allele. Thus, it is considered that resistant DRB3*009:02 allele can strongly suppress the development of high PVL than DRB3*014:01:01 allele or DRB3*002:01 allele. It may be that DRB3*009:02 allele combines with other factors to thoroughly suppress high PVL. In addition, R cattle have lower PVL and weak BLV infectivity than susceptible cattle, indicating that R cattle are low risk of BLV transmission. We have added this information in line 301-308 in the revised manuscript.

Some literature has reported that DRB3 allele associations with BLV resistance and susceptibility differ between cattle breeds, and that allele may be associated with resistance to one aspect of BLV (e.g. PVL) but not others (e.g., lymphoma; see e.g., Brujeni 2016 doi: 10.1007/s10528-016-9712-6). A curiosity question follows from this line of thinking – do resistant and susceptible alleles to one – or multiple –aspect(s) of BLV disease ever occur in the same animal, and if so, what implications or consequences might that have for BLV control programs?

Answer: We agree with the reviewer’s comment, two BoLA-DRB3*010:01 and DRB3*011:01 alleles were specifically associated with lymphoma resistance, but no lymphoma-specific susceptibility alleles were found; furthermore, two other alleles, DRB3*002:01 and DRB3*012:01, were associated with PVL resistance and susceptibility, respectively. In contrast, lymphoma and PVL shared two resistance-associated DRB3*014:01:01 and DRB3*009:02 alleles (Lo, C.W. et al., Viruses 2020, 12, 352; doi:10.3390/v12030352). Some cattle carry one susceptible DRB3 allele and one resistant DRB3 allele in their genome. In this case, most susceptibility alleles are more dominant and usually classify as susceptible cattle.

I agree with the authors that genetic considerations could be important for BLV control programs. However, I think it is important to clearly acknowledge that a breeding strategy for BLV control requires consideration of multiple other elements, given associations between genetics and key factors in the dairy industry such as milk production and reproductive efficiency (see, e.g., Abdalla 2016 doi:10.3168/jds.2015-9833).

Answer: Thank you so much for your comments.

As mentioned in the overall summary, I think it is particularly important to be clear about when the authors are 1) making statements regarding their study versus reporting findings from literature, and 2) discussing BLV infection (the state of having BLV), infectivity, or transmission/transmission risk. I have indicated some specific examples below along with other minor comments.

  • Line 271: Here is an example where I think the authors may mean “infectivity” rather than infection.

Answer: As reviewer suggestion, we have changed “infection” to “infectivity” in line 276 in the revised manuscript.

  • Line 292-293: suggest “differ significantly” – the current tense implies that this was examined in the current study.

Answer: As reviewer suggestion, we have modified to “differ significantly” in this sentence in line 317 in the revised manuscript.

  • Line 311-312: I am not sure the conclusion that resistant cattle are not a source of infection (in vivo) is supported by this study. I suggest framing this in terms of risk, similar to the next sentence.

Answer: As reviewer suggestion, we have modified “…are not a source of infection…” to “…are low risk of BLV transmission…” in this sentence in line 335-336 the revised manuscript.

  • Line 315: Here again, I wonder if the authors are talking about infectivity or becoming BLV infected.

Answer: As reviewer suggestion, we have changed “BLV infection” to “BLV infectivity” in this sentence in line 339 in the revised manuscript.

  • Line 321: insert “the”

Answer: As reviewer suggestion, we have inserted “the” in line 345 in the revised manuscript.

Materials and Methods

The Materials and Methods are light on detail, for me, particularly given that this is a genetic study. Additional information would be useful, including selection criteria and herd demographics (broadly). For example:

  • Is farm “A” a commercial farm? Dairy? Beef? What is the breed composition of the herd?

Answer: As reviewer suggestion, we have added more detail information “In Japan, the number of cattle raised per farm is 93.9. We selected an entire herd of 179 cattle containing Holstein cattle, Japanese Black, and F1 hybrids, which is about twice as large as the average scale” about the selected farm in line 348-350 in the revised manuscript.

  • The Results section 2.1 indicates that the entire herd of 179 cattle were sampled – this is good information that I would expect to see in the Materials and Methods.

Answer: As response in above reviewer comment, we have added more detail information about the selected farm A in line 348-350 in the revised manuscript.

  • In Results section 2.2, the authors indicate that 24 cattle were tested using LuSIA. What were the selection criteria for these 24 cattle? In addition, it appears that only a subset of these samples was further analyzed which raises the question of why this is the case and the selection criteria for this additional analysis.

Answer: In response to reviewer comment, we randomly tested 24 cattle from 142 BLV-infected cattle for selecting out the samples for follow-up study. Through the BLV infection experiment of 24 cattle in vitro, we have obtained that the BLV infectivity of resistant cattle is weaker than the susceptible cattle. Therefore, we decided to use syncytium formation assay to clarify the difference of BLV infectivity between susceptible cattle and resistant cattle.

   In this three-year follow-up investigation, we would not constrain any decisions of the farmer as depositary breeding, selling, slaughtering, and eliminating et al. At the beginning, we decided to select 13 out of 24 cattle for infectivity syncytium formation assay and correlation analysis of BLV infectivity with PVL, but some selected cattle were sold out or deposited with other vendors' deposit breeding services, while at the time of our sampling, so only 7 out of 13 cattle were sampled at each checking point. Therefore, only 7 samples were enrolled for infectivity syncytium formation assay. When the cattle in the depositary breeding service returned to the farm, they were sampled again for correlation analysis of BLV infectivity with PVL. We have added this information in line 198-205 in the revised manuscript.

  • Similarly, what were the selection criteria for the 7 cows in the three-year follow-up study and why was it limited to a total of 7 cattle?

Answer: As response in above reviewer comment, we would not constrain any decisions of the farmer as depositary breeding, selling, slaughtering, and eliminating et al. in this three-year follow-up investigation. At the beginning, we decided to select 13 out of 24 cattle for infectivity syncytium formation assay and correlation analysis of BLV infectivity with PVL, but some selected cattle were sold out or deposited with other vendors' deposit breeding services, while at the time of our sampling, so only 7 out of 13 cattle were sampled at each checking point. Therefore, only 7 samples were enrolled for infectivity syncytium formation assay. When the cattle in the depositary breeding service returned to the farm, they were sampled again for correlation analysis of BLV infectivity with PVL. We have added this information in line 198-205 in the revised manuscript.

  • Were any statistical power calculations performed when determining what seem to be small group sizes? If not, I feel this should be acknowledged because it impacts the interpretation and generalizability of the results.

Answer: In response to reviewer comment, we did not do any statistical calculations. Juliarena M.A. et al. has reported resistant cattle disrupted the BLV-transmission chain that BLV-infected cattle carrying resistant BoLA-DRB3*009:02 allele were incorporated into a BLV-negative dairy herd and no BLV-negative cattle became infected after 30 months of contact (J. Dairy Sci. 2015, 99, 4586-4589). In this three-year follow-up study, resistant cattle had weak BLV infectivity and low PVL than susceptible cattle, which is consistent with other report. Therefore, resistant cattle are low risk of BLV transmission. We have added this information in “Discussion” section in Line 284-288 in the revised manuscript.

A suggestion I would also make is to include classification definitions (susceptible, resistant, etc.) in the methods section, though that may be a matter of preference since these definitions are stated in the Results section.

Answer: As reviewer’s suggestion, we have added the definitions of susceptible and resistant in “Methods” section in line 372-375 in the revised manuscript.

Conclusions

I feel that the authors are trying to cover a lot of ground in this paragraph while keeping it brief, which has resulted in some confusing statements. In particular, I had trouble comprehending the sentence beginning in Line 376 (see below).

  • Line 373: remove “with”

Answer: As reviewer suggestion, we have removed “with” from the sentence in line 401-403 in the revised manuscript.

  • Line 376: I had trouble comprehending this sentence. I think the authors are saying that the number of cattle in each group that was analyzed was not equal (i.e., the groups were of different sizes) but I’m a little unsure.

Answer: In response to reviewer comment, we apologize that our sentence makes the reviewer confusing. We did not have samples taken from susceptible cattle and resistant cattle that had same amount of PVL at each checking point. Therefore, we cannot directly compare the BLV infectivity between susceptible cattle and resistant cattle had same amount of PVL. Whereas resistant cattle keep weak infectivity and low PVL in long infection period. In here, we want to explain the above information as “Although we did not directly compare BLV infectivity in the same amount of PVL of susceptible cattle and resistant cattle in vitro, the BoLA-DRB3 allele was found to be directly or indirectly associated with BLV infectivity” in the submitted manuscript.

  • Line 378: Later in the same sentence, I again am uncertain what the authors are saying regarding BLV infectivity in vivo – I understood this study to include in vitro assays only. This may be another case where clarity regarding BLV-infection, infectivity, or transmission would help.

Answer: As reviewer suggestion, we have deleted “in vivo” from the sentence in line 407 in the revised manuscript.

In line with my comments on the Abstract, I think it is an overstatement of the study results (which did not measure in vivo transmission) to state that genetic selection on this allele is an effective strategy. In addition, as mentioned above, potentially limiting elements of this study (e.g., some group sizes appearing to be quite small, using a single herd of (one might assume) one or two cattle breeds, and so on) may affect the generalizability of conclusions drawn from the study results. However, this study adds valuable information to the growing body of evidence for important genetic effects on BLV infectivity and supports genetic selection as a potentially useful strategy for BLV control (see, e.g., Gutiérrez 2017, doi: 10.5772/65804).

Answer: Thank you for your comment. In Japan, the number of cattle raised per farm is 93.9. We selected an entire herd of 179 cattle containing Holstein cattle, Japanese Black, and F1 hybrids, which is about twice as large as the average size. We think that more different breeds and big scale samples to confirm our result in next study.

Round 2

Reviewer 1 Report

Authors had addressed most of my comments.

Author Response

Pathogens

Author's Response To Reviewer Comments

Answer to the comments of reviewer #1:

Comments and Suggestions for Authors

Authors had addressed most of my comments.

Answer: Thank you so much for the helpful and constructive comments.

Reviewer 3 Report

Thank you for your good work on this revised manuscript. Although I am still of the opinion that the blanket statement that “breed[ing] selection based on resistant BoLA-DRB3 allele is an effective strategy for reducing and controlling BLV infection” (Line 407-408) is an overstatement of the conclusions that can be drawn from this study, most of my major concerns have been addressed. My remaining comments on this revision are primarily related to clarity and English language writing. The additions to the Discussion, in particular, could use some revision – again mostly for clarity rather than content.

Specifically:

The paragraph starting at Line 277 is quite long and covers a few major topics. I recommend grouping these topics together in their own paragraphs because your message is quite strong, and I do not want it to be lost. For example, this paragraph starts with a focus on the *009:02 allele as a potent allele for resistance. I suggest to you that you could move the sentence about other reports on the stability of PVL over time (Line 290) to Line 287 and wrap up this paragraph with the sentence about resistant cattle presenting a low risk of BLV transmission. You could then explore your discussion of other, less potent resistance alleles in the next paragraph, and then discuss your hypotheses about the mechanisms of resistance alleles in a subsequent paragraph.

Line 284-287:  suggest re-wording for clarity; I believe the authors are intending to say that, in the referenced study, no BLV transmission was observed over 30 months when BLV+ cattle carrying the resistance allele *009:02 [and, I will note, which were known to have low PVL] were introduced into a BLV- herd.

Line 287-8:  suggest re-wording here also, e.g., Thus, resistant cattle with the *009:02 allele can be considered low risk for transmitting BLV.

Line 288-294:  I think these sentences were added to address my concerns about generalizability, which is appreciated; however, I think these statements would be better used elsewhere (see my Discussion comments above) to maintain the flow of the Discussion and because the topic here is not related to generalizability. I would suggest you either add a paragraph to address generalizability specifically, or omit the sentence at Line 288 (Furthermore, our selected farm A…). The statement about the size of the farm is included in the Methods after your revisions, and the distribution of resistant/susceptible is included in the results, so that information remains available in the manuscript if the latter option is chosen.

Line 290:  insert ‘are’ and correct tense of reported – “Our results are consistent with previous reports…”

Line 292:  please include citations to the referred to previous reports

Line 293:  am I correct to assume this statement refers to the observation in this study? This could be more clear.

Author Response

Pathogens

Author's Response To Reviewer Comments

Answer to the comments of reviewer #3:

Thank you very much for the helpful and constructive comments. We have amended the manuscript in accordance with these comments and our point-by point responses are set out below.

Comments and Suggestions for Authors

Thank you for your good work on this revised manuscript. Although I am still of the opinion that the blanket statement that “breed[ing] selection based on resistant BoLA-DRB3 allele is an effective strategy for reducing and controlling BLV infection” (Line 407-408) is an overstatement of the conclusions that can be drawn from this study, most of my major concerns have been addressed. My remaining comments on this revision are primarily related to clarity and English language writing. The additions to the Discussion, in particular, could use some revision – again mostly for clarity rather than content.

Answer: In response to reviewer comment, cattle carrying resistant BoLA-DRB3 allele have lower PVL and weaker infectivity comparing with other cattle, indicating that resistant cattle are at low risk of BLV transmission. Therefore, we mentioned as “breed selection based on resistant BoLA-DRB3 allele is an effective strategy for reducing and controlling BLV infection” in line 413-415 in this revised manuscript.

   In response to reviewer comment, we have separated our long paragraph in “Discussion” to short paragraphs to clarity. Furthermore, we have added a new paragraph to explain generalizability in line 304-310. In addition, we have modified our English language in this revised manuscript.

Specifically:

The paragraph starting at Line 277 is quite long and covers a few major topics. I recommend grouping these topics together in their own paragraphs because your message is quite strong, and I do not want it to be lost. For example, this paragraph starts with a focus on the *009:02 allele as a potent allele for resistance. I suggest to you that you could move the sentence about other reports on the stability of PVL over time (Line 290) to Line 287 and wrap up this paragraph with the sentence about resistant cattle presenting a low risk of BLV transmission. You could then explore your discussion of other, less potent resistance alleles in the next paragraph, and then discuss your hypotheses about the mechanisms of resistance alleles in a subsequent paragraph.

Answer: Thank you so much for your recommendation, we have moved the sentence about other reports on the stability of PVL over time (Line 290) to Line 287 and wrap up this paragraph with the sentence about resistant cattle presenting a low risk of BLV transmission in this revised manuscript.

   As reviewer suggestion, we have separated this long paragraph starting at Line 277 to short paragraphs at line 277, line 291, line 304, and line 313 in this manuscript.

Line 284-287:  suggest re-wording for clarity; I believe the authors are intending to say that, in the referenced study, no BLV transmission was observed over 30 months when BLV+ cattle carrying the resistance allele *009:02 [and, I will note, which were known to have low PVL] were introduced into a BLV- herd.

Answer: As reviewer comment, we have modified our sentence as “no BLV transmission was observed over 30 months of contact when cattle carrying resistant BoLA-DRB3*009:02 allele with low PVL were incorporated into a BLV-negative dairy herd. The BoLA-DRB3*0902 cattle with low PVL disrupted the BLV transmission chain” in line 284-287 in this revised manuscript.

Line 287-8:  suggest re-wording here also, e.g., Thus, resistant cattle with the *009:02 allele can be considered low risk for transmitting BLV.

Answer: As response the above comment, we have modified our English language in these sunstones line 284-287 in this revised manuscript.

Line 288-294:  I think these sentences were added to address my concerns about generalizability, which is appreciated; however, I think these statements would be better used elsewhere (see my Discussion comments above) to maintain the flow of the Discussion and because the topic here is not related to generalizability. I would suggest you either add a paragraph to address generalizability specifically, or omit the sentence at Line 288 (Furthermore, our selected farm A…). The statement about the size of the farm is included in the Methods after your revisions, and the distribution of resistant/susceptible is included in the results, so that information remains available in the manuscript if the latter option is chosen.

Answer: As reviewer comment, we have moved the sentence at line 288 as “Furthermore, our selected farm A…” from this revised manuscript. We have newly added a paragraph as “In Japanese Black herds, the distribution of allele frequencies for BoLA-DRB3*016:01 was over 30%, DRB3*009:02 was 2.0 ~ 5.8%, and DRB3* 014:01:01 was 1.4 ~ 2.5%. In Holstein cattle herds, the distribution of allele frequencies for BoLA-DRB3*015:01 was 19.4~ 35.4%, DRB3*009:02 was 5.2 ~7.1%, DRB3* 014:01:01 was 2.4 ~ 7.1%, and DRB3*002:01 was 4.5 ~ 8.6% (Taku Miyasaka, Gene, 2011, 472, 42-49). The distribution of BoLA-DRB3 allele frequencies in our selected farm A also was in the above distribution ranges. Thus, farm A can be a representative of Japanese farms, and it can interpret Japanese farms” to explain generalizability in line 304-310 in “Discussion” section in this revised manuscript

Line 290:  insert ‘are’ and correct tense of reported – “Our results are consistent with previous reports…”

Answer: As reviewer points out, we have inserted the word “are” into “Our results are consistent with previous reports…” in line 287 in this revised manuscript.

Line 292:  please include citations to the referred to previous reports.

Answer: As reviewer points out, we have added a new reference 48 and reference 33 in line 289 in this manuscript.  

Line 293:  am I correct to assume this statement refers to the observation in this study? This could be more clear.

Answer: As reviewer points out, the statement at Line 293 refers to the observation in this study. Therefore, we have modified to “Therefore, our results are considered that can reflect the state of the entire farm” in line 311 in the revised manuscript.
